# Spontaneous Confinement of mRNA Molecules at Biomolecular Condensate Boundaries

**DOI:** 10.3390/cells12182250

**Published:** 2023-09-11

**Authors:** Rebecca T. Perelman, Andreas Schmidt, Umar Khan, Nils G. Walter

**Affiliations:** 1Single Molecule Analysis Group, University of Michigan, Ann Arbor, MI 48109, USA; btperel@umich.edu (R.T.P.); andreasschmidt58@gmail.com (A.S.); 2Center for Advanced Biomedical Imaging and Photonics, Beth Israel Deaconess Medical Center, Harvard University, Boston, MA 02115, USA; ukhan1@bidmc.harvard.edu; 3Center for RNA Biomedicine, Department of Chemistry, University of Michigan, Ann Arbor, MI 48109, USA

**Keywords:** membraneless organelles, mRNA, HILO microscopy, liquid–liquid phase separation, biomolecular condensates, RNP granules, colocalization, intermolecular interactions, Donnan potential

## Abstract

Cellular biomolecular condensates, termed ribonucleoprotein (RNP) granules, are often enriched in messenger RNA (mRNA) molecules relative to the surrounding cytoplasm. Yet, the spatial localization and diffusion of mRNAs in close proximity to phase separated RNP granules are not well understood. In this study, we performed single-molecule fluorescence imaging experiments of mRNAs in live cells in the presence of two types of RNP granules, stress granules (SGs) and processing bodies (PBs), which are distinct in their molecular composition and function. We developed a photobleaching- and noise-corrected colocalization imaging algorithm that was employed to determine the accurate positions of individual mRNAs relative to the granule’s boundaries. We found that mRNAs are often localized at granule boundaries, an observation consistent with recently published data. We suggest that mRNA molecules become spontaneously confined at the RNP granule boundary similar to the adsorption of polymer molecules at liquid–liquid interfaces, which is observed in various technological and biological processes. We also suggest that this confinement could be due to a combination of intermolecular interactions associated with, first, the screening of a portion of the RNP granule interface by the polymer and, second, electrostatic interactions due to a strong electric field induced by a Donnan potential generated across the thin interface.

## 1. Introduction

Biomolecular condensates are membraneless compartments in eukaryotic cells that most often contain RNA-binding proteins and mRNA molecules in concentrations higher than the surrounding cytoplasm [1]. Ribonucleoprotein (RNP) granules, a class of biomolecular condensates containing significant portions of RNA-binding proteins [2], have been found to play critical roles in epigenetic and post-transcriptional regulation [3] and, consequently, have been associated with numerous diseases, such as tumor progression and neurodegeneration [4,5]. RNP granules form liquid- or solid-like compartments of RNA and proteins [6] through processes, including liquid–liquid phase separation (LLPS) and percolation [7]. 

RNP granules often contain messenger RNAs (mRNAs) in concentrations higher than the surrounding cytoplasm. At the same time, the spatial localization and dynamics of mRNAs in close proximity to RNP granules are not well understood. We here present imaging data obtained with low-background highly inclined and laminated optical (HILO) sheet microscopy in live cells containing two types of RNP granules: stress granules (SG) and P-bodies (PB). We developed a coregistration imaging algorithm to determine the exact positions of the mRNAs relative to the granule boundaries, which revealed that ~80% of colocalized mRNAs were located at the SG and PB boundaries. These observations are independently supported by recent studies showing that mRNAs are predominantly found at the boundaries of biomolecular condensates [4,8,9]. 

A number of mechanisms, such as multivalent interactions [10] and protein condensation [11], often induced by cellular stress [2,12,13], could be responsible for forming the RNP granule interface. In turn, these mechanisms may potentially play a role in the confinement of mRNA at the interface. However, many mechanisms, such as multivalent, hydrophobic, or amphiphilic interactions, strongly depend on the specific chemical composition of the RNP granule and, therefore, may not be applicable to every type of granule [14,15]. For this boundary localization with both SGs and PBs to occur for a majority of mRNAs, which can be either translating or nontranslating [8], we posit that a universal mechanism is at work. We speculate that it may resemble the well-known surface adsorption of polymer molecules, driven by the strong electric fields induced by the Donnan potential across the thin granule-cytoplasm interface.

## 2. Materials and Methods

### 2.1. Cell Line Cultivation, Handling, and Imaging

U2-OS (HTB-96, ATCC) cells stably expressing GFP-G3BP1 (UGG cells) were maintained in DMEM, whereas U2-OS-GFP-DCP1A (UGD) cells were maintained in McCoy’s 5A medium (GIBCO). Both cells were supplemented with 10% (*v*/*v*) fetal bovine serum (GIBCO) and 1% (*w*/*v*) penicillin–streptomycin (GIBCO) at 37 °C under 5% CO_2_. UGD cells were kept under positive selection with 100 μg/mL G418 [13]. U2-OS cells stably expressing GFP-G3BP1 were a gift from the Moon lab [4]. Oxidative stress was induced in UGG cells by treating them with 0.5 mM sodium arsenite (NaAsO_2_, SA) for 60 min to form SGs. 

Messenger RNA was loaded into cells via bead loading [16]. Briefly, the medium was removed, and the cells were washed twice with 1 mL 1× PBS buffer; then, 5 µL of RNA solution in 1× PBS buffer (200 ng) was added to the center of the glass dish, followed by an addition of glass beads. Next, the cell dish containing the glass beads was tapped 10 times against the bench, the culture medium was added back and incubated at 37 °C for 60 min (SG experiments) and 75 min (PB experiments) before imaging. For the live cell imaging, the cells were imaged in phenol-red free Leibovitz’s L-15 medium containing 1% FBS using a Nanoimager S microscope from Oxford Nanoimaging Limited (ONI) in the highly inclined laminated optical sheet (HILO) illumination mode [17]. Alexa Fluor 647 dye and GFP were detected using 640 nm and 473 nm lasers, respectively. The camera integration time was 100 ms. 

### 2.2. Generation of Fluorophore-Labeled mRNA

The synthesis of fluorophore-labeled mRNA included four steps [18]. First, an enzymatic run-off in vitro transcription was performed using T7 RNA Polymerase (ThermoFisher Scientific, Waltham, MA, USA), as described by the manufacturer. DNA transcription templates were generated from a corresponding plasmid using PCR. A total of 5% (*v*/*v*) of the unpurified PCR product was used as a template for the in vitro transcription at 37 °C for 90 min. Following the ethanol precipitation, the RNA was resuspended in the double-deionized water. Second, the 3′-end was azide-functionalized using yeast poly(A) polymerase (yPAP, ThermoFisher Scientific) in the presence of 2′-azido-2′-deoxyadenosine-5′triphosphate (ATP-azide, Trilink Biotechnologies, San Diego, CA, USA) and capped using the ScriptCap m7G Capping System (CellScript). The one-pot reaction mixture contained 1.4 µM of RNA transcript, 600 µM SAM, 1 mM GTP, 700 µM ATP-azide, 10 U capping enzyme, 2400 U of yPAP and 1× Script capping buffer. Following an incubation at 37 °C for 60 min, the reaction mixture was ethanol precipitated and resuspended in double-deionized water. Third, the recovered RNA was polyadenylated in the presence of ATP using 2400 U yeast poly(A) polymerase (yPAP, ThermoFisher Scientific) in 1× poly(A) polymerase reaction buffer at 37 °C for 30 min. Fourth, in the same mix fluorophore labeling was achieved by adding a 150-fold molar excess of Click-IT Alexa Fluor 647 sDIBO Alkyne (ThermoFisher Scientific) at room temperature for 60 min, followed by another ethanol precipitation. On average, it is estimated that 23 to 27 fluorophore labels were attached to each mRNA molecule [18]. 

### 2.3. Image Sequence Colocalization

The acquired image sequence consisted of the HILO microscopy-based hyperstacks of images with one of the channels containing mRNA images and the other containing granule images. The images were processed using image sequence colocalization with the histogram matching particle detection algorithm we developed. The data processing steps included a photobleaching correction with histogram matching, noise removal with progressive switching median filters, Laplacian of Gaussian-based particle detection, and colocalization of all detected and isolated granules and mRNA molecules. 

The channel containing the single mRNA molecule images exhibited a significant decay in signal strength over time due to the gradual photobleaching of the multiple fluorophores per mRNA. To compensate for this decrease, we employed a histogram matching transformation [19,20]. Histogram matching involves the adjustment of pixels of an unprocessed image or sequence of images to match them with a reference image. This is performed by finding the cumulative histogram of both the reference and original image. Then, the cumulative histogram of the original unprocessed image(s) is adjusted to match the unprocessed image(s) with the cumulative distribution function of a chosen reference image. Figure 1 provides an example of the histogram matching transformation performed on one of the frames from our image sequence. The correction for photobleaching can be evidenced for an image at 10 s that lacks contrast; the histogram matching algorithm corrects the contrast by using the reference image at 0 s, which results in a high-contrast image at 10 s.

To detect mRNA and granules and obtain a granule’s area in the image sequence dataset, we implemented the scale-normalized Laplacian of Gaussian (LoG) detection algorithm, which is well suited for our application because of its strong response to particles residing in inhomogeneous background [21]. The pixel size of our image acquisition was 134 nm in real space. An mRNA was determined to be colocalized with a granule if the coordinates of the mRNA were within ±134 nm of the boundary, as determined using the LoG algorithm, that is, colocalized within the diffraction limit of the microscope. The algorithm then characterized it as located at the boundary and presented it as a green dot with its coordinates located at the centroid of the mRNA point spread function (see Appendix A). The remaining colocalized mRNAs are presented as red dots, whereas noncolocalized mRNAs are presented as blue dots.

### 2.4. Statistical Analyses

Statistical details such as the number of mRNAs analyzed are indicated in the text and figure legends.

## 3. Results

### 3.1. Visualization of mRNA and RNP Granules in Live Cells with HILO Microscopy

Two U2-OS cell lines were used, each one stably transfected with GFP-G3BP1 (termed UGG cells) and with GFP-Dcp1a (termed UGD cells) to label SGs and PBs, respectively [9,22]. The cells were treated with 500 μM sodium arsenate to induce oxidative stress and form SGs [9,22]. We enzymatically synthesized a 1600 nt long, 5′-capped, poly-adenylated, and Cy5-labeled mRNA [16] and introduced it into both cell types via bead loading [23]. The cells were then placed onto a glass bottom cell dish and incubated for 45 min in the case of UGG cells and for 60 min in the case of UGD cells. During incubation, SGs and PBs formed while single mRNA molecules diffused throughout the cells. Additional information on cell and mRNA types is provided in Section 2.

We then performed single-molecule mRNA imaging and RNP granule visualization via HILO microscopy [17] on a Nanoimager (ONI). The simultaneous acquisition of GFP and Cy5 fluorescence emitted by the granule and mRNA labels, respectively, was performed in two coregistered channels with the emitted light split via a beamsplitter onto two halves of a high-sensitivity camera (Hamamatsu sCMOS Orca flash 4 V3) [24]. The resulting hyperstacks of images contained mRNA images in one channel and the granule images in the other. The cells were kept at 37 °C using an onstage incubator. The detectors’ pixel size reflected 134 nm in real space, and the microscope image sequences were collected every 100 ms for 20 s (200 images in total). Each imaging experiment was replicated once.

Representative images from a time series of a UGG cell with SGs overlaid on the channel with mRNAs is presented in Figure 2a and Appendix A. The bottom panels show part of a continuous 20 s long image time series of the mRNA, which appears to be confined at the boundary of the SG. The mRNA follows the SG movements, demonstrating that the mRNA is, in fact, positioned on the granule boundary and not outside of the granule or attached to a nearby structure (see Appendix A). A similar behavior can be observed in UGD cells, where the mRNAs are located in the proximity of PBs. A representative image time series of a UGD cell with PBs and mRNAs is shown in Figure 2b and Appendix A. Again, the mRNA appears to be localized at the boundary of the PB for the duration of the observation (see Appendix A). 

### 3.2. Colocalization of the Single mRNA Molecule with an RNP Granule Boundary

Because of photobleaching, the duration of the microscopy image sequences was limited to 20 s, over which the fluorescence signal of the single mRNA molecules dropped noticeably. To address this challenge, we developed an algorithm that performs photobleaching correction using a histogram matching method [19,20], described in Section 2. The image sequence colocalization with the histogram matching particle detection algorithm was implemented with ImageJ [25]. In addition to photobleaching correction, we reduced the noise in the images with progressive switching median filters and performed RNP granule detection, which were treated as extended objects, using the Laplacian of Gaussian (LoG) particle detection method [26]. Since the size of single mRNA molecules is significantly smaller than the 134 nm pixel resolution of the camera detector, they were treated as points. As such, because of diffraction, the mRNA images were blurred to a diffraction limited spot. To overcome the diffraction blurring, we used the fact that because of the intentionally sparse spatial distribution of the mRNA molecules in our images, the chances of mRNAs overlapping are exceedingly low. Consequently, the signals in the single-molecule detection channel originated mostly from individual, nonoverlapping mRNAs. As a result, the coordinates of the centroid of the blur, obtained using the algorithm described in [27], provide the coordinates of the mRNA from frame to frame, with interpolation allowing for subdiffraction accuracy in single particle tracking. An mRNA was considered colocalized with a granule if the mRNA coordinates were within the area of the RNP granule determined with the LoG method and/or if the mRNA was located within ±134 nm from the granule boundary. In the latter case, the mRNA was considered both colocalized and detected at the boundary within the diffraction limits of the microscopy system. We also computed and stored the distances among all mRNAs located within two radii from the granules and the granules’ boundaries.

In addition to the quantitative description of the mRNAs and RNP granules, the algorithm allowed for the processing of a large number of microscopy stacks where manual processing would be rather difficult and would risk introducing bias.

Typical outputs of mRNA colocalization with the RNP granules boundaries are presented in Figure 3 where two live UGG cells containing SGs (Figure 3a) and two UGD cells containing PBs (Figure 3b) are visualized. The algorithm determined the locations of the boundaries of the granules using the scale-normalized Laplacian of Gaussian (LoG) detection algorithm (see Section 2). The boundaries were assigned but not shown in order to improve the visibility of the images and because the locations of the boundaries are quite obvious because of the sharp drop in intensity at the edges of the granules. Out of 14 mRNAs colocalized with the RNP granules in these images, 12 were within ±134 nm, that is, within one diffraction limit from the granule boundary.

The algorithm determined the mRNA centroid coordinates from the single-molecule HILO microscopy images and marked the locations on the cell images in color. The size of the dots does not represent the size of the mRNA molecules, which was assumed to be much smaller than the pixel size but instead arbitrarily chosen only for visualization purposes. The centers of the dot, coinciding with the maximum fluorescence intensities of the mRNA labels, are the centroids of the mRNAs. Each mRNAs whose centroid coordinates were inside or on the boundary of the RNP granules are shown as red and green dots, respectively, and the remainder of the mRNAs are shown as blue dots. Furthermore, all of the mRNAs colocalized with SGs in Figure 3a are shown zoomed in on the side of the original image. It appears that in this particular image, out of the nine colocalized mRNAs, seven were located within the diffraction limited distance from the boundary of the granule and, therefore, on the boundary, whereas two mRNAs were inside the SG. A similar observation holds for the PBs in Figure 3b, where all five colocalized mRNAs were located on the boundary.

### 3.3. Spatial Density Distribution of mRNA Molecules in Stress Granules and Processing Bodies

Figure 3 was chosen to be representative of all of our data, where mRNAs were predominantly found at the RNP granule boundaries in both UGD and UGG cells. Overall, we determined that out of ~10,000 mRNA molecules that we imaged in the UGD and UGG cells, around 80% were located on the boundaries of the PBs and SGs. Furthermore, as the microscopy system has a finite depth of view, some mRNA molecules that are actually located on the boundary of a granule may appear to be localized to the inside of the granule in a given microscopy images. Therefore, these percentages represent a lower estimate of the fraction of mRNAs that are localized to the granule boundaries. 

To assess this finding more quantitatively, we determined the average spatial density distributions of mRNAs relative to the boundary of the nearest RNP granule, with the distance measured to the closest point of the granule (Figure 4). Negative distances represent individual mRNA locations outside the granule, whereas positive distances represent mRNAs observed inside of the granule. For both PBs and SGs, a significant peak of mRNA density appears at the boundaries of the granules, indicating that mRNA molecules often become spontaneously confined specifically at the RNP granule boundary. These observations are generally consistent with data from other recent studies that similarly observed mRNA molecules predominantly at the boundaries of biomolecular condensates [4,8,9]. For example, in recent work combining imaging in the visible wavelength range with confocal fluorescence microscopy of mRNAs in SGs, over two-thirds of the mRNAs were located, within the optical resolution limit, at the granules’ boundaries [8].

### 3.4. Fast and Slow Dwell Times of mRNA 

Other more indirect observations also support our findings. For example, recently it was discovered that mRNA imaged in close proximity to RNP granules are best described with at least two characteristic dwell times [4]: a fast time on the order of seconds [28,29,30,31] and a slow time where an mRNA becomes “stuck” to the RNP granule for tens of minutes and even hours [4]. It is thought that the shorter of these dwell times is likely related to the diffusion of mRNAs within the RNP granule. By contrast, the nature of the characteristic long dwell time is less clear, but it is consistent with our observation of a fixed boundary localization that lasts for at least 20 s (Figure 2). One potential explanation for the existence of two characteristic times may be related to the known inhomogeneity of RNP granules [32]. However, such an inhomogeneity would be expected to result in multiple time constants associated with the mRNA traveling across dense cores of various sizes and with the various distances among these cores [4]. Additionally, one would expect these times to differ among different types of biomolecular condensates. Therefore, it seems more probable that the long characteristic time is associated with the spontaneous confinement of the mRNA at the RNP granule interface, reflecting the majority of our mRNA molecules.

To assess further whether this confinement relates to the fast or slow dwell time, we estimated the characteristic time, *τ_d_*, of an mRNA molecule diffusing across an RNP granule of size *d*. The diffusion of mRNA can be described using Brownian motion in a highly viscous liquid. From Einstein’s expression for the mean squared displacement in three dimensions, $\langle r^{2}\rangle =6D\tau_{d}$, where *D* is the diffusion coefficient of mRNA in an RNP granule, and assuming that on average $\sqrt{\langle r^{2}\rangle }\approx d/2$, using the standard Einstein–Stokes equation for the diffusion coefficient, we obtain $\tau_{d}\approx \pi d^{2}\eta R_{g}/4k_{B}T$, where *R_g_* is the radius of gyration of mRNA, *T* is the temperature, and *k_B_* is the Boltzmann constant. The majority of the parameters in this equation vary within a rather narrow range, making the viscosity, *η*, of the liquid- to solid-like medium inside the RNP granule the single most important parameter to affect the characteristic diffusion time of an mRNA molecule within an RNP granule. Experimental measurements in several types of biomolecular condensates, including P granules [33], TDP-43 RNP granules [34], PGL-3 condensates [35], and condensates formed in LAF-1 [33,34,35,36] have yielded viscosities in the range of 0.1 to 34 Pa·s. Even for the highest viscosity in that range at room temperature, for a typical mRNA with a radius of gyration of 10 nm and a relatively large one-micrometer granule, the characteristic diffusion time would be approximately two minutes. This is an order of magnitude shorter than the 20 to 60 min previously observed experimentally [4] and would entail the single mRNA visibly diffusing within the granule. Therefore, in order to associate the slow time with the diffusion of an mRNA within an RNP granule, the viscosity of the granule would have to be at least 400 Pa·s or, at least, an order of magnitude higher than observed. 

Taken together, we posit that a boundary association mechanism not related to Brownian diffusion causes the long characteristic dwell time observed previously [4] and here. It would appear that spontaneous confinement of individual mRNA molecules at the RNP granule boundary occurs based on an interaction with the liquid–liquid interface. 

## 4. Discussion

We here used single-molecule fluorescence imaging of mRNAs in live cells harboring SGs or PBs, two well-known and functionally relevant cellular types of RNP granules [1,6,7], and developed an algorithm for the photobleaching- and noise-corrected extraction of accurate mRNA positions over time relative to the granule boundaries (Figure 1). We found that individual mRNA molecules predominantly (to ~80% in our experimental setting) become localized to granule boundaries, independent of the type of granule (Figure 2, Figure 3 and Figure 4). In the following, we discuss plausible mechanisms.

We used two types of RNP granules, SGs and PBs, of distinct biochemical makeup. Furthermore, the mRNA used in our studies is different from the mRNAs employed in Refs. [4,8,9], where the data similarly show that mRNAs were predominantly found at the boundaries of biomolecular condensates. Taken together, these observations suggest that the confinement of mRNA molecules to biomolecular condensate boundaries is universal and mostly independent of the makeup of the biomolecular condensate and the type of RNA.

mRNAs are highly polyanionic molecules. Polymers generally can experience confinement or adsorption at the interface of two liquids. This phenomenon has been extensively studied experimentally, theoretically, and computationally [37,38,39]. The desorption time (equivalent to the dwell time) is a parameter that characterizes the degree of polymeric molecule confinement and can be measured experimentally. The desorption time of a single polymer molecule from a liquid–liquid interface has also recently been modeled computationally [40]. One plausible mechanism for polymer adsorption involves the molecular-level amphiphilicity of the polymer. However, a more general explanation is related to the fact that the presence of a polymer molecule at the liquid–liquid interface can locally lower the interfacial free energy, trapping the polymer molecule at the interface [41].

In addition to the change in the surface-free energy of the liquid–liquid interface due a portion of it being screened by the polymer molecule, the change in the free energy upon desorption depends on the change in the polymer interaction energy and conformational entropy when the molecule is moved from the interface into the bulk liquid. However, for the desorption from liquid–liquid interfaces, changes associated with the polymer conformational entropy represent only a small fraction of the total free energy of the system [42] and can, therefore, be safely neglected. Similarly, computer simulations indicate [43] that the contribution of the interaction energy of the polymer molecule with both the bulk cytoplasm, which is a poor solvent for RNA and DNA [44], and the interface can also be neglected. Therefore, based on what is known about polymer adsorption and desorption at a liquid–liquid interface, intermolecular interactions associated with the screening of a portion of the interface by the polymer are most likely to play a role in the experimentally observed spontaneous confinement of mRNA molecules at RNP granule boundaries.

Examples for experimental observations of intermolecular interaction effects acting at biomolecular condensate interfaces are the trapping of clusters of RNA-binding protein MEG-3 [35] and of microtubules [45]. In the case of individual macromolecules, such as an mRNA, the situation is somewhat complicated by the fact that lowering the local interfacial free energy of the system, which entails a decrease in the liquid–liquid interfacial area in the vicinity of the polymer, is often accompanied by conformational changes such as a flattening of the polymer [46]. 

To describe intermolecular interactions of a polymer molecule at the biomolecular condensate interface, the polymer molecule can be treated as a chain of “blobs” [47] with an effective spring constant for blob-to-blob interactions being weak compared to the force that the interface exerts on the blob. That is, the interaction between blobs should not noticeably affect the desorption time. Previously published experimental results, observed mostly for solid-liquid interfaces, suggest that the polymer is confined at the interface by a single train segment [48]. In the case of a chain of blobs, this structure could be a portion of a single blob residing within the interface. Figure 5 provides a conceptual interpretation of this scenario for the case of an mRNA molecule confined by the RNP granule liquid–liquid interface.

Another type of interaction that may contribute to the observed confinement of mRNAs at the RNP granule boundary is electrostatic in nature. It has been shown earlier that a strong electric field is induced across a membrane in membrane-bound organelles such as mitochondria [49]. We propose that a similarly strong electric field might be induced across a phase separation interface such as an RNP granule boundary. For electrically conducting liquids, phase separation requires electrochemical potentials, rather than the chemical potentials, to be equal on both sides of the interface for each species. This feature is called the Donnan equilibrium and results in an electrostatic potential difference across the phase separation interface. It can be observed in mixtures of electrolytes where the low-mobility phase, containing higher concentrations of large ions, is permeable to small highly mobile ions [50]. Such conditions are present in eukaryotic cells where biomolecular condensates are formed not only by electroneutral molecules of the cytosol and RNP granules, but also small mobile ions, such as potassium, sodium, chloride, bicarbonate, magnesium, and calcium, as well as amino acids, metabolites, and charged proteins and nucleic acids, at various concentrations. Even though the Donnan potential is relatively small (less than 8 mV [51]), it is generated across a very thin interface, approximately one Debye in length, which leads to a generated field as high as 1–10 V/μm. This is just an order of magnitude lower than the field strength of the dielectric breakdown of cytosol, which is comparable to that of water [52]. 

For a biological macromolecule present in a cell to interact with this electric field at the biomolecular condensate interface, the macromolecule does not have to be charged. In fact, it can be electroneutral with its positive and negative charges completely compensating one another. Instead, the macromolecule needs to have a nonzero permanent or induced mean square dipole moment. The induced dipole moment of RNA for strong fields is, at least, an order of magnitude larger than their permanent dipole moment [53]. Therefore, the interaction of the induced dipole moment of mRNA with the electric field induced by the Donnan potential is likely to contribute to the observed trapping of mRNA molecules at the granule boundary. We are currently working on evaluating the characteristic confinement times due to the change in the liquid–liquid interface surface-free energy and the electric field induced by the Donnan potential.

## 5. Conclusions

In conclusion, we report the first observation of the confinement of single mRNA molecules at the cytoplasmic boundary of two types of RNP granules: SGs and PBs. We posit that this effect is common among many types of biomolecular condensates. We further suggest that this spontaneous confinement of polyanionic mRNAs is similar in nature to the surface adsorption of polymer molecules observed in various technological and biological processes. Future studies can now test how general mRNA confinement at the interfaces of other types of biomolecular condensates is and further establish which interaction mechanism is primarily responsible for the confinement.

## Figures and Tables

**Figure 1 cells-12-02250-f001:**
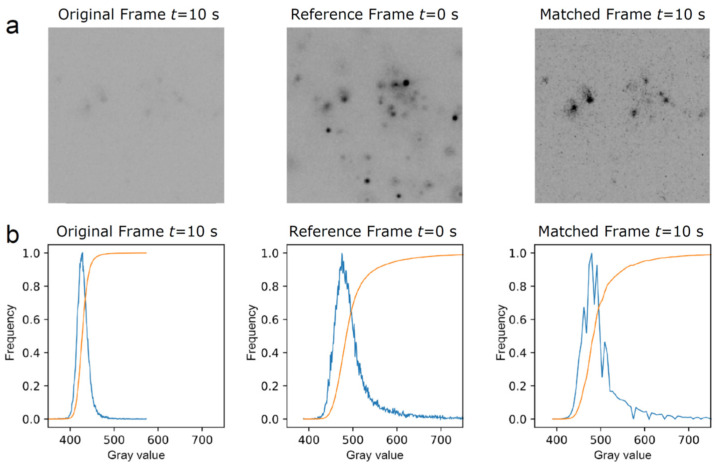
Example illustrating histogram matching with gray value adjustment of a given frame in an image sequence using the first frame as a reference. (**a**) The original frame, taken at 10 s, is low in contrast, whereas the revised matched frame at 10 s shows an improved contrast after its histogram is matched with the reference frame at 0 s. (**b**) Plots derived from the respective frames above showing histograms (in blue) and cumulative distribution functions (in orange). Here, the gray value represents the intensity of gray in a pixel, with the horizontal axis representing the gray values present in the frame and the vertical axis representing the frequency of occurrence of these gray values.

**Figure 2 cells-12-02250-f002:**
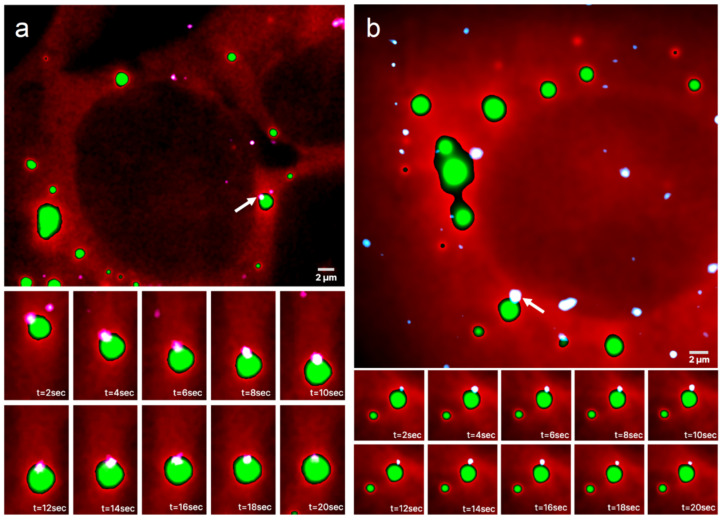
Time series of mRNAs in live cells containing GFP-labeled RNP granules. Representative HILO microscopy images of a (**a**) UGG cell (red) with GFP-labeled SGs (green) overlaid with the channel visualizing mRNAs (white/light pink) and a (**b**) UGD cell (red) with GFP-labeled PBs (green) overlaid with the channel visualizing mRNAs (white/light blue). The bottom panels show a 20 s long time series of the magnified area containing an mRNA molecule, marked with a white arrow in the top panel, that appears to be confined to the boundary of the (**a**) SG and (**b**) PB. The mRNA in (**a**) follows all of the SG movements.

**Figure 3 cells-12-02250-f003:**
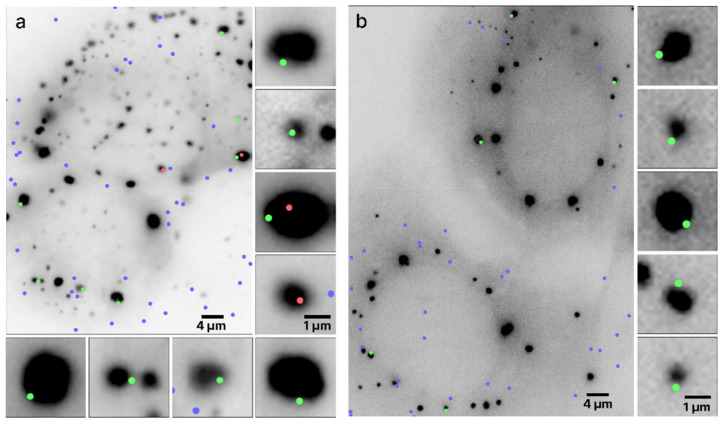
Colocalization of mRNAs and RNP granules. Two live UGG cells containing (**a**) SGs and two live UGD cells containing (**b**) PBs with granules rendered as dark structures. The dots represent the locations of the mRNA centroids interpolated with subdiffraction accuracy. The green and red dots represent mRNA molecules colocalized with the RNP granules, with the green dots specifically representing mRNAs whose centroid coordinates are within ±134 nm from the boundary. The blue dots represent any mRNAs not colocalized with the RNP granules. Specific mRNAs colocalized with PBs and SGs are shown zoomed in on the right and bottom of the respective panels, approximately directly across from their location in the cell.

**Figure 4 cells-12-02250-f004:**
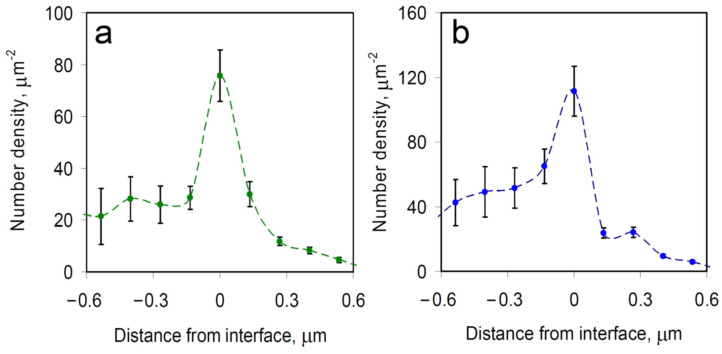
Average spatial density distribution of single mRNA molecules relative to the boundaries of the nearest (**a**) SG and (**b**) PB. Negative distances represent locations outside of the granule, whereas positive distances represent locations on the inside, with 0 μm denoting the location of the boundary. The results represent the means ± SEM.

**Figure 5 cells-12-02250-f005:**
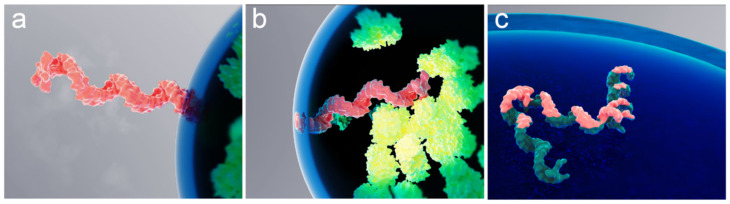
Rendering of various spontaneous confinements of a single mRNA molecule at an RNP granule interface. Schematic interpretation of the mRNA molecule (pink) trapped at the interface by a “single blob”, with the rest of the molecule located either (**a**) outside or (**b**) inside of the granule. Also shown is a molecule “trapped” by its entire (**c**) length. Proteins inside the granule are depicted in green.

## Data Availability

The data that support the findings of this study are available from the corresponding author upon reasonable request.

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
