# Peer review of "Spontaneous Confinement of mRNA Molecules at Biomolecular Condensate Boundaries"

_cells, 2023, doi:10.3390/cells12182250_

Round 1
Reviewer 1 Report
The authors performed single molecule fluorescence imaging of mRNAs on RNP granules (stress granules and processing bodies) and observed that most of the mRNA localize to granule boundaries using a new algorism to determine accurate positions of individual mRNA. The authors state that this principal observation was already published by several other researchers including the corresponding author himself what questions the novelty of the data and added value to the research community.
The authors suggest a universal mechanism based on electrostatic fields on the surface of the granules which confine mRNA. The experiments proving this suggestion are ongoing. These experiments may clearly elucidated the mechanisms of mRNA interactions with SG granules and P-bodies and may add scientific value to the manuscript.
The present manuscript is structured in a rather technical/methodical manner and the conclusions are mainly based on previously published data. It should therefore be considered whether a review article on the dynamics of biomolecular condensates with mRNA supplemented by the improved image acquisition algorithm used here would be more useful.
Specific comments:
Fig. 2
In figure 2b no pink label to see as described in the figure legend? The arrow indicates to a white label and there are blue, green and red signals to see.
No control with RNP granules with different electrostatic surface (different amino acid sequence) and different Cy5-labeled RNA species or molecules to control specificity of confinement at the boundaries or inside the granules.
Figure. 3
No indication from which part of the cell the enlarged images are taken! Original mRNA labeled image must be shown with the color dot labeled mRNA image.
Movie showing mRNAs confinement to the granules only in the supplement mentioned?
Author Response
Referee #1
- The authors performed single molecule fluorescence imaging of mRNAs on RNPA granules (stress granules and processing bodies) and observed that most of the mRNA localize to granule boundaries using a new algorithm to determine accurate positions of individual mRNA The authors state that this principle observation was already published by several other researchers including the corresponding author himself what questions the novelty of the data and added value to the research community.
We apologize that we apparently did not relay clearly that our paper is the first to recognize that mRNAs are confined to biomolecular condensate boundaries. We cited three publications, including one previously published by our group, wherein the data corroborate our work to show that mRNAs are concentrated predominantly near the condensate boundary. In actuality, none of those papers reported the confinement. We have now clarified this in the revised manuscript (line 48).
- The conclusions are mainly based on previously published data.
Once again, we apologize for not making it clear that the conclusions of the manuscript are based on our own data, and not on previously published data. On page 7, the manuscript states that approximately 10,000 mRNAs were imaged in the UGD and UGG cells using single-molecule HILO microscopy. We have now clarified this further in the revised manuscript (line 246).
- It should be therefore be considered whether a review article on the dynamics of the biomolecular condensates with the mRNA supplemented by the improved image acquisition algorithm used here would be more useful.
The suggestion to consider turning the paper into a review article apparently stems from the misunderstanding addressed in the answers to concerns 1 and 2. The manuscript reports a new and original result and is based on our original data. We have now clarified this further in the revised manuscript (line 404).
- In Figure 2b no pink label to see as described in figure legend? The arrow indicates to a white label and there are blue, green and red signals to see.
We agree with the referee that the choice of language describing the colors in Figure 2 is confusing. We have now corrected the caption to clarify the choice of colors (lines 178-179).
- No control with RNP granules with different electrostatic surface (different amino acid sequence) and different Cy5-labeled RNA species or molecules to control specificity of confinement at the boundaries or inside the granules.
Indeed, demonstrating that the confinement of mRNA to biomolecular condensate boundaries is a sufficiently universal effect is an important part of our manuscript. For this reason, we used two types of RNP granules, SGs and PBs, which have a different biochemical makeup. Furthermore, the mRNA used in our studies is different from the mRNAs employed in Moon et al. Nat. Cell Biol. [4], Mateju et al. Cell [8] and Pitchiaya et al. Mol. Cell. [9], where the data similarly show that mRNAs are predominantly found at the boundaries of biomolecular condensates. Notably, these prior works do not comment on this feature observed in their imaging. Therefore, we feel that we can conclude with confidence that the confinement can be observed for RNP granules and mRNAs of different composition, and that we are the first to state so. Still, we agree that this point was not sufficiently clearly articulated in the manuscript and we have now clarified in in the revision (lines 319-325).
- Figure 3: No indication from which part of the cell the enlarged images are taken! Original mRNA labeled image must be shown with the color dot labeled mRNA image.
In the Figure 3 caption, we state that “Specific mRNAs colocalized with PBs and SGs are shown zoomed in on the right and bottom of the respective panels, approximately directly across from their location in the cell.” Showing the original mRNA labeled image instead of dots representing centroids of the blur would not improve the visibility since the mRNAs will appear at exactly the same locations where the centroid dots are, but will be blurred over a larger number of pixels due the diffraction limit, partially blocking PBs and SGs. The blurring due to the diffraction limit mRNAs can be observed in Figure 2.
- Movie showing mRNAs confinement to the granules only the supplement method?
Thank you for pointing this out, we have now referred to the movies in the main text of the manuscript (lines 166, 170, and 172-174).

Reviewer 2 Report
Biomolecular condensates are a distinct type of membraneless cellular structure, originating from phase separation, responsible for executing a wide range of biological functions. In their research, Perelman et al. utilized single-molecule tracking technology to study mRNA localization in stress granules and processing bodies. Their observations reveal that mRNAs predominantly localize to boundary areas, suggesting an inherent confinement of RNA at liquid-liquid interfaces. The manuscript is cogently organized and effectively communicates its main points. This reviewer suggests adding a detailed workflow diagram that outlines the imaging and processing algorithm, further elucidating the data processing steps.
Author Response
Referee #2
- This reviewer suggests adding a detailed workflow diagram that outlines the imaging and processing algorithm, further elucidating the data processing steps.
We agree with the referee that providing such a workflow diagram will be helpful for better understanding the results of the manuscript, so we have now provided such a diagram as a new Figure S1 and refer to it in the main text (lines 138 - 139, and 413-414).

Reviewer 3 Report
The article: “Spontaneous Confinement of mRNA molecules at Biomolecular Condensate Boundaries” describes the nature of biomolecular condensates enriched with RNA-binding proteins and mRNA molecules with an emphasis on the imagine algorithms for determination of spatial localization and dynamics of mRNAs in close proximity of RNP stress granules (SG) and processing bodies (PB).
The entire manuscript is well written, with sufficient illustrative and data explanation in the Materials and Method section and the Result section. The obtained data are adequately discussed in light of available data from the corresponding research field, and the manuscript is complemented with a related and up-to-date reference list.
Supplementary data notifications should be incorporated in the manuscript text.
Other than that, acceptance of the manuscript and its publication in the current form is suggested.
Author Response
Referee #3
- Supplementary data notifications should be incorporated in the manuscript text.
We apologize for not including supplementary data notifications in the main text of the manuscript; we now refer to the movies and the new supplementary figure in the main text of the manuscript (lines 138-139, 166, 170, 172-174, and 413-414).
